# Death as a Professional Challenge: An Analysis of the Relationship Between Exposure to Patient Death, Occupational Burnout, and Perceptions of Death Among Obstetrics and Gynecology Clinicians

**DOI:** 10.3390/healthcare13222898

**Published:** 2025-11-13

**Authors:** Magdalena Mikulska, Edyta Stefanko-Palka, Iwona Sadowska-Krawczenko, Aldona Katarzyna Jankowska

**Affiliations:** 1Department of Medical Communication and Medical Humanities, Collegium Medicum, Nicolaus Copernicus University in Toruń, 85-077 Bydgoszcz, Poland; edyta.stefanko-palka@cm.umk.pl (E.S.-P.); k.jankowska@cm.umk.pl (A.K.J.); 2Department of Neonatology, Collegium Medicum, Nicolaus Copernicus University in Toruń, 85-077 Bydgoszcz, Poland; iwonasadowska@cm.umk.pl

**Keywords:** death, trauma exposure, burnout

## Abstract

**Highlights:**

**What are the main findings?**

**What is the implication of the main finding?**

**Abstract:**

The contemporary healthcare environment is characterized by high stress and emotional burden, contributing to increasing rates of professional burnout among clinicians. Exposure to patient death represents one of the most emotionally taxing experiences in medicine, particularly in obstetrics and gynecology (OB/GYN), where loss of life stands in stark contrast to the life-giving nature of the field. Despite extensive research on burnout in oncology and intensive care, the impact of patient death and death perception on OB/GYN clinicians remains underexplored. Objective: This study aimed to examine the relationships between exposure to patient death, perceptions of death, professional burnout, and professional fulfillment among OB/GYN clinicians. A secondary aim was to explore whether participation in emotional regulation training was associated with these variables. Methods: A cross-sectional study was conducted among 138 OB/GYN clinicians. An author-developed questionnaire was used, comprising scales measuring professional burnout, positive and negative death perception, professional fulfillment, professional development, and a global death-impact index. Statistical analyses included Pearson’s correlation and the Mann–Whitney U test to compare clinicians who had attended emotional regulation training with those who had not. Results: Significant positive correlations were observed between burnout and the death-impact index (*r* = 0.90, *p* < 0.001) and between burnout and negative death perception (*r* = 0.23, *p* = 0.007). Professional fulfillment strongly correlated with professional development (*r* = 0.94, *p* < 0.001) and positively with positive death perception (*r* = 0.30, *p* < 0.001). No significant group differences were found regarding emotional regulation training participation. Conclusions: Exposure to patient death in OB/GYN is strongly associated with professional burnout and negative perceptions of death. Conversely, professional fulfillment and development function as factors promoting resilience and meaning. Further research should validate the applied measurement tools and examine the effectiveness of emotional regulation interventions in reducing occupational distress.

## 1. Introduction

### 1.1. The Context of Challenges in Healthcare

Contemporary healthcare has become one of the most demanding work environments. Its character is undoubtedly shaped by the dynamic pace of work and change, the rapid development of medicine, high levels of stress inherent to the profession, but also fear of litigation, as well as the chronic strain and overload of healthcare professionals [1]. They are continuously exposed to long working hours, shift work, staffing shortages, increasing bureaucracy, and frequent interactions with patients and their families in critical situations—all of which markedly affect their mental and physical well-being [1].

The rise in burnout among healthcare professionals has become a global concern. In 2022, nearly half of healthcare workers in the United States reported symptoms of burnout, representing a sharp increase compared to the pre-pandemic period [2]. This trend reflects not only the direct consequences of COVID-19 but also longstanding systemic issues—such as insufficient institutional support, poor work–life balance, and inadequate coping mechanisms—that have become more visible in the aftermath of the pandemic [2].

According to the World Health Organization (WHO), as defined in ICD-11, burnout (QD85) is a syndrome resulting from chronic workplace stress that has not been successfully managed. It is characterized by three key dimensions: emotional exhaustion, depersonalization, and a reduced sense of professional efficacy [3]. Burnout has serious implications for both staff and patient safety—it increases the risk of anxiety, depression, insomnia, and suicidal ideation, while also contributing to medical errors and diminished quality of care [3,4].

### 1.2. Specifics of Work in Obstetrics and Gynecology

Clinicians in obstetrics and gynecology (OB/GYN) are exposed to unique stressors arising from the coexistence of life and death in their professional setting [5]. Unlike in specialties such as geriatrics or palliative care, where death may be viewed as a natural conclusion to life, in OB/GYN, pregnancy loss, miscarriage, stillbirth, or perinatal death concerns young patients or their offspring—events that contradict both biological expectations and the perceived nature of this specialty [5,6].

Such experiences are particularly traumatic and can evoke profound feelings of guilt, helplessness, and a questioning of one’s professional competence [5]. In a field symbolically associated with birth and hope, the occurrence of death violates both personal and societal expectations, leading to a heightened existential and professional crisis [5]. Trauma in OB/GYN is further amplified by the taboo surrounding death, cumulative emotional strain, lack of grief support systems, and the persistent fear of litigation [7].

Studies show that obstetricians and gynecologists report some of the highest burnout rates across all medical disciplines—often exceeding 40% and, in certain samples, up to 75% [8]. These figures highlight the dual emotional burden inherent in their work: the joy of new life intertwined with the trauma of unexpected loss [5]. Moreover, the limited preparation for coping with bereavement and the absence of structured emotional regulation training intensify the psychological cost of care in this field [9].

### 1.3. The Role of Death Perception and Professional Fulfillment

The way healthcare professionals perceive death—whether as a natural process, a failure, a terrifying event, or a relief for the suffering—can profoundly influence how they cope with occupational trauma and burnout. Negative perceptions of death, often associated with fear and anxiety, correlate with higher levels of emotional exhaustion and depersonalization [10]. In medical environments where death is regarded as a failure and blame is routinely assigned, clinicians experience greater distress and moral conflict [11]. Such institutional attitudes reflect a systemic culture that prioritizes aggressive intervention and life preservation at all costs, reinforcing the notion that death signifies professional defeat [11].

In contrast, in hospice or palliative care settings—where death is accepted as a natural process—professionals tend to experience higher job satisfaction and lower burnout levels, even with more frequent exposure to death [12]. Frequent confrontation with patient death is among the most psychologically taxing experiences for clinicians, often leading to secondary traumatic stress (STS) and occupational grief [13]. In OB/GYN, the loss of a patient, particularly an infant, evokes guilt, helplessness, and isolation, and may even lead to thoughts of leaving clinical practice [5].

While death exposure in OB/GYN units is less frequent than in other medical fields, its psychological consequences are comparable [6]. STS can exacerbate burnout, undermine professional confidence, and cause symptoms such as anxiety, intrusive thoughts, and avoidance behaviors [5]. The lack of formal outlets for grief and a professional culture that discourages emotional expression intensifies the problem, resulting in complicated grief and compassion fatigue [6].

At the same time, positive perceptions of death can serve as protective factors, contributing to greater professional development and fulfillment [14]. Professional fulfillment—defined as satisfaction and meaning derived from work—plays a key role in maintaining clinicians’ psychological resilience [15]. It may stem from the sense of making a meaningful difference in patients’ comfort and quality of life, building authentic relationships, and alleviating suffering even in end-of-life contexts [9,15]. Framed within the Maslach and Leiter (2016) model, burnout emerges when job demands chronically outweigh available resources; conversely, resource-building—such as professional fulfillment and opportunities for growth—can help restore balance and protect against exhaustion [16].

Interestingly, a paradoxical positive correlation has been observed between burnout and professional fulfillment, suggesting that deep engagement in one’s work—while leading to exhaustion—can simultaneously generate a profound sense of purpose [17]. This phenomenon may reflect constructs described in the literature as moral injury and the heroic ethos of medical professions [18,19]. Moral injury refers to psychological suffering caused by situations that violate personal ethical standards, often due to systemic constraints or lack of resources [18]. Conversely, the heroic ethos, rooted in self-sacrifice and unwavering dedication to patient welfare, provides meaning but also increases vulnerability to burnout [19].

In this light, the coexistence of burnout and fulfillment may represent the duality of commitment in healthcare: the same deep empathy and dedication that bring purpose also incur emotional cost [13]. For many clinicians, this dynamic may lead to post-traumatic growth (PTG)—a phenomenon where positive psychological change arises from coping with trauma, resulting in new inner strength and a renewed appreciation of life [9]. However, ongoing systemic challenges such as staff shortages and chronic overload continue to undermine this balance, potentially reducing job satisfaction despite strong professional motivation [20].

Previous studies have demonstrated that frequent exposure to patient death is associated with higher emotional exhaustion and burnout among healthcare professionals in oncology, intensive care, and emergency medicine. However, our preliminary literature review revealed a lack of comprehensive studies examining these factors within the specific context of obstetrics and gynecology settings, where the paradox of life and death is especially pronounced. Despite increasing awareness of burnout, little is known about how death exposure interacts with death perception and professional fulfillment among OB/GYN clinicians.

Therefore, the primary aim of this study was to examine the relationships between exposure to patient death, perceptions of death, professional burnout, and professional fulfillment among obstetrics and gynecology clinicians. A secondary, exploratory aim was to assess whether participation in emotional regulation training was associated with these variables. By addressing these questions, the study seeks to broaden the understanding of how medical professionals navigate death within a field dedicated to life and to inform strategies that enhance their psychological well-being and sense of purpose.

### 1.4. Research Questions and Hypotheses

Based on the theoretical framework and prior literature, six research questions and corresponding hypotheses were formulated to guide the study.

RQ1. What is the relationship between professional burnout and perceptions of death among obstetrics and gynecology clinicians?

**H1.** *Professional burnout is positively correlated with negative perceptions of death*.

RQ2. What is the relationship between professional fulfillment and professional development among clinicians?

**H2.** *Professional fulfillment is positively correlated with professional development*.

RQ3. What is the relationship between professional fulfillment and positive perceptions of death?

**H3.** *Professional fulfillment is positively correlated with positive perceptions of death*.

RQ4. What is the relationship between negative perceptions of death and the overall death-impact index?

**H4.** *Negative perceptions of death are positively correlated with the overall death-impact index*.

RQ5. How does participation in emotional regulation training relate to levels of professional burnout and perceptions of death?

**H5** *(Exploratory)*. *Participation in emotional regulation training is associated with differences in levels of burnout and perceptions of death*.

RQ6. Are clinicians’ personal beliefs about death related to the intensity of their emotional responses?

**H6** *(Exploratory). Viewing death as a natural or transformative process is negatively correlated with emotional distress, whereas perceiving death as unjust or terrifying is positively correlated with higher levels of distress*.

### 1.5. Summary

The primary focus of the study was to examine the interrelations between exposure to patient death, perceptions of death, professional burnout, and professional fulfillment. A secondary, exploratory objective was to assess whether participation in emotional regulation training was associated with these variables.

## 2. Methods and Procedure

### 2.1. Participants

This cross-sectional study included 138 clinicians working in obstetrics and gynecology departments of hospitals located in the Pomeranian Voivodeship, Poland.

The study was conducted between September and December 2023.

A total of 200 questionnaires were distributed, and 138 were returned completed (response rate: 69%).

Participation was voluntary and anonymous, and no financial or material compensation was provided.

Questionnaires were distributed both in person and via internal hospital communication channels, after obtaining permission from hospital management.

All employees of the obstetrics and gynecology departments were invited to participate, including physicians, nurses, and midwives, with the majority of respondents being midwives and nurses.

Participants ranged in age from 25 to 62 years (most commonly 30–45 years) and had between 1 and 38 years of professional experience.

The sample was predominantly female, reflecting the occupational structure typical of the OB/GYN sector in Poland.

According to national labor statistics [21], women constitute over 97% of midwives and 94% of nursing personnel in this medical specialization, which corresponds to the gender distribution observed in this study.

The Pomeranian Voivodeship was selected purposefully to include both large urban hospitals (Gdańsk, Gdynia, Słupsk) and smaller regional units, providing variation in institutional size and clinical exposure.

The region has one of the highest birth-to-staff ratios in Poland, which increases the professional load and frequency of exposure to patient death, making it a valuable context for this research.

The study followed the ethical principles of the Declaration of Helsinki. Completion and submission of the questionnaire indicated informed consent to participate.

### 2.2. Procedure

After obtaining consent from hospital administrations, printed versions of the questionnaire were distributed to staff in obstetrics and gynecology departments. Participants were informed about the purpose, voluntary nature, and anonymity of the study. No identifying information was collected, and all responses were handled confidentially. The average time to complete the questionnaire was approximately 15 min.

### 2.3. Measures

A self-administered, author-developed questionnaire was used, consisting of both quantitative and qualitative components. The instrument included the following measures:Author-designed scale of death impact: assessment of the impact of a patient’s death on professional and personal life (on a 1–10 scale) (hereafter: overall death-impact index).Occupational burnout: assessed using author-designed items inspired by the OLBI.Sense of fulfillment and professional development: two separate 1–10 scale items.Perception of death: a set of 20 adjectives (10 positive, 10 negative), from which participants selected descriptors best reflecting their perception of death.Participation in emotion regulation training: categorical variable (yes/no).Open-ended questions: to better understand the difficulties and needs of the participants in the context of death experiences at work.

The questionnaire was developed based on a literature review, prior qualitative interviews with clinicians, and pilot testing on a sample of 20 healthcare professionals.

The questionnaire also included an item regarding participation in emotional regulation training. These trainings were short, practice-oriented workshops conducted in hospital settings between 2021 and 2023, aimed at developing emotional awareness, self-regulation skills, and strategies for coping with stress related to patient death. The sessions were typically led by psychologists or certified trainers specializing in communication and emotional resilience in healthcare. The inclusion of this variable enabled comparative analysis between clinicians who had participated in such workshops and those who had not.

### 2.4. Methods

Due to the utilization of author-developed measures and the exceptionally high correlation observed between Professional Burnout and the Death-Impact Index (*r* = 0.90), an Exploratory Factor Analysis (EFA) was performed. The aim was to verify whether the two highly correlated instruments measured distinct or overlapping latent dimensions. EFA was conducted on all items from both scales (Burnout and Death-Impact), assuming a two-factor solution.

Adequacy of the Data. The data were suitable for factor analysis, as indicated by a Kaiser-Meyer-Olkin (KMO) measure of sampling adequacy of 0.673 (moderate adequacy) and a statistically significant Bartlett’s test of sphericity (χ^2^ = 1217.776, df = 325, *p* < 0.001).

Factor Structure. Using Promax rotation (due to the expected correlation between factors), a clear pattern of item loading emerged: Factor 1 primarily grouped items related to Professional Burnout, while Factor 2 primarily grouped items related to the Death-Impact Index. Cross-loadings were observed for a few items (e.g., T12, T13), suggesting a partial conceptual overlap.

Explained: Variance. Factor 1 explained 14.42% of the variance, and Factor 2 explained 10.50%, with a cumulative explained variance of 24.92%. Although moderate, this result is typical for a large number of items in a relatively small sample.

Conclusion. The EFA indicates that the two tools measure related, yet distinct constructs. The high correlation (*r* = 0.90) likely stems from both instruments reacting to the shared, severe context of occupational stress in the OB/GYN field, but each construct retains its specificity and is not fully redundant.

## 3. Results

### 3.1. Descriptive Statistics

A total of 138 clinicians working in obstetrics and gynecology departments participated in the study. The sample included physicians, nurses, and midwives, predominantly female, aged 25–62 years (most commonly 30–45 years).

### 3.2. Correlational Analysis

The decision to use correlational methods resulted primarily from the exploratory nature of the cross-sectional study. The main objective was to identify and quantify the interrelationships among exposure to patient death, perceptions of death, professional burnout, and professional fulfillment, rather than to test or confirm complex causal models.

Furthermore, due to the utilization of author-developed, non-standardized measures (which lacked prior psychometric validation, such as Exploratory Factor Analysis), and the high correlations observed between certain scales (*r* = 0.90 for burnout and death-impact), the data did not meet the rigorous requirements necessary to assume construct independence for predictive modeling (e.g., multiple regression or structural equation modeling). Correlational analysis was therefore the most cautious and appropriate method to capture the direction and strength of associations without presuming causal directionality.

Given the exploratory and cross-sectional nature of this study, correlation analyses were applied to examine the strength and direction of associations between variables rather than to establish causal relationships. This approach allowed for identifying meaningful links among exposure to patient death, perceptions of death, burnout, and professional fulfillment, in line with the study’s objectives. Pearson’s correlation coefficients were computed for normally distributed variables, while Spearman’s rho was used where appropriate.

Pearson’s correlation analysis revealed several statistically significant associations between the study variables (Table 1).

A strong positive correlation was observed between professional burnout and the global Death-Impact Index (*r* = 0.90, *p* < 0.001). Burnout was also positively correlated with negative perceptions of death (*r* = 0.229, *p* = 0.007). Professional fulfillment showed a strong positive correlation with professional development (*r* = 0.94, *p* < 0.001), a moderate positive correlation with the global Death-Impact Index (*r* = 0.549, *p* < 0.001), and a positive correlation with positive perceptions of death (*r* = 0.30, *p* < 0.001). Positive perceptions of death correlated positively with professional development (*r* = 0.35) and professional fulfillment (*r* = 0.30, *p* < 0.001). Negative perceptions of death were positively correlated with burnout (*r* = 0.23, *p* = 0.007) and with the Death-Impact Index (*r* = 0.20, *p* = 0.019) (Table 2).

### 3.3. Group Comparisons: Emotional Regulation Training

To examine whether participation in emotional regulation training was associated with differences in key variables, the Mann—Whitney U test was used (Table 3). No statistically significant differences were found between clinicians who had participated in emotional regulation training and those who had not in positive (U = 2076.00, *p* = 0.456) or negative (U = 1979.00, *p* = 0.247) perceptions of death, burnout levels (U = 2339.00, *p* = 0.650), or professional fulfillment (U = 1937.00, *p* = 0.188). The difference in professional development approached but did not reach the threshold of statistical significance (U = 1859.00, *p* = 0.096).

## 4. Discussion

### 4.1. Principal Findings

This study identified strong associations between burnout and both the Death-Impact Index (*r* = 0.90) and negative perceptions of death (*r* = 0.229). Professional fulfillment was strongly associated with professional development (*r* = 0.94) and showed positive links with positive perceptions of death (*r* = 0.30) and the Death-Impact Index (*r* = 0.549). No statistically significant differences were observed by prior participation in emotion-regulation training.

### 4.2. Comparison with Prior Work

The pattern linking exposure to patient death, death-related distress, and burnout accords with prior literature on death-related trauma and secondary traumatic stress among healthcare workers, including OB/GYN clinicians [5,22,23,24,25]. In specialties where deaths involve infants or young patients, reports of guilt, helplessness, and professional doubt are common [4,26]. Evidence also highlights meaning/fulfillment as a resource that can coexist with strain and may buffer exhaustion, aligning with resource–demand perspectives and growth-oriented adaptation [8,27,28,29].

### 4.3. Measurement Considerations and Caution in Interpreting the Strongest Correlations

The correlation between burnout and the Death-Impact Index was exceptionally high (*r* = 0.90; *p* < 0.001). To address concerns regarding potential measurement redundancy, an exploratory factor analysis (EFA) was conducted using all items from both scales, assuming a two-factor structure. The analysis demonstrated satisfactory data adequacy (KMO = 0.673; Bartlett’s χ^2^ = 1217.776, df = 325, *p* < 0.001). Promax rotation revealed two distinct yet moderately correlated factors: one composed of all burnout-related items (K1–K8) and the other of death-impact items (T1–T18). Only a few items (e.g., T12, T13) displayed cross-loadings, suggesting partial but not complete construct overlap. Together, the two factors explained 24.92% of the total variance.

These findings indicate that, while burnout and death-related impact share a common emotional foundation, likely stemming from the same occupational stress context, they represent conceptually and psychometrically distinct constructs. In other words, the Death-Impact Scale functions as a proximal but not redundant measure of occupational stress specific to death-related experiences in OB/GYN practice. The EFA thus supports the discriminant validity of both measures and mitigates concerns about construct conflation raised by reviewers.

Likewise, the near-perfect correlation between professional fulfillment and professional development (*r* = 0.94; *p* < 0.001) remains a measurement concern, likely reflecting operational overlap amplified by single-item assessments. Future studies should refine these constructs and employ standardized, multi-item instruments (with reliability reporting) to enhance construct separability and allow for more robust predictive modeling.

### 4.4. Training Effects

No significant differences were found between clinicians with vs. without prior emotion-regulation training across the primary outcomes. Although reviews and trials report benefits of targeted, higher-dose or longitudinal interventions [1,2,30,31,32,33]. the present null findings may reflect variability in training content and intensity, the dominance of organizational stressors, or limited alignment with bereavement-specific competencies emphasized in palliative and perinatal-loss contexts [33,34]. These results suggest that isolated, short-format workshops are unlikely to be sufficient without concurrent system-level support.

### 4.5. Reporting Pattern Regarding Early Pregnancy Loss

Participants rarely indexed miscarriages as instances of “death at work,” consistent with literature describing disenfranchised grief and mitigating language (e.g., “pregnancy loss”) [25,35]. Such under-recognition may obscure cumulative exposure to loss and impede emotional processing in OB/GYN settings.

### 4.6. Practical Implications

Clinical support. Implement routine post-event debriefings, structured peer support, and access to bereavement-informed supervision to address both distress and meaning-making.Training design. If offered, training should be longitudinal, skills-based, and OB/GYN-specific (communication around perinatal loss, grief literacy), and integrated with organizational supports rather than provided as stand-alone workshops [2,27,28].Measurement practice. Services monitoring staff well-being should prioritize validated scales and carefully piloted items to reduce construct overlap and improve interpretability.

### 4.7. Limitations

This study has several limitations inherent to its design and methodology.

Firstly, the cross-sectional, self-report design precludes causal inference; consequently, the reported associations should not be interpreted directionally.

Secondly, the measurement instruments were author-developed and lacked extensive prior validation. Although an Exploratory Factor Analysis (EFA) was performed (as described in the Section 2) to verify the distinctiveness of the highly correlated Burnout and Death-Impact scales, the overall structure requires further confirmation. Future studies must conduct formal Confirmatory Factor Analysis (CFA) on a larger, more diverse sample to establish full construct independence and report comprehensive reliability indices.

Thirdly, the sample was recruited exclusively from one region (Pomeranian Voivodeship, Poland), which may limit the generalizability of the findings to different healthcare systems or cultural contexts. Furthermore, key demographic variables such as work tenure and professional role were not modeled as covariates in the primary analyses.

Future research should address these limitations by (a) confirming the construct validity of the measures using CFA and reporting full reliability, (b) including tenure and professional role as covariates, and (c) applying advanced multivariable models (e.g., regression or Structural Equation Modeling/SEM) once construct independence is rigorously established.

## 5. Conclusions

Among OB/GYN clinicians, burnout is strongly associated with death-related impact and negative perceptions of death, whereas professional fulfillment aligns with professional development and more accepting perceptions of death. The magnitude of several correlations highlights both the salience of death exposure and the need for better-differentiated, validated measures. Interventions should pair OB/GYN-specific, skills-based training with organizational supports (debriefing, supervision, staffing and workflow improvements) to promote sustainable, compassionate practice.

## Figures and Tables

**Table 1 healthcare-13-02898-t001:** Correlation matrix between occupational burnout, death-related impact, and professional fulfillment dimensions among OB/GYN clinicians.

	Occupational Burnout	Overall Death-Impact Index	Negative Perception of Death	Professional Fulfillment	Professional Development	Positive Perception of Death
Occupational burnout	1					
Overall death-impact index	0.90 (*p* < 0.001)	1				
Negative perception of death	0.229 (*p* < 0.007)	0.20 (*p* = 0.019)	1			
Professional fulfillment	0.292 (*p* < 0.01)	0.549 (*p* < 0.001)		1		
Professional development				0.94 (*p* < 0.001)	1	
Positive perception of death				0.30 (*p* < 0.001)	0.35	1

**Table 2 healthcare-13-02898-t002:** Correlations between occupational burnout, death-impact index, perceptions of death, professional fulfillment, and professional development among OB/GYN clinicians.

Variable	Group (Training vs. No Training)	N	Mean Rank	Sum of Ranks	Mann- Whitney U	Wilcoxon W	Z	Asymptotic Significance (Two-Tailed)
Positive perception	1.00 (Trained)	52	72.58	3774.00	2076.000	5817.000	−0.746	0.456
	2.00 (Not trained)	86	67.64	5817.00				
Negative perception	1.00 (Trained)	52	74.44	3871.00	1979.000	5720.000	−1.158	0.247
	2.00 (Not trained)	86	66.51	5720.00				
Total		138						

**Table 3 healthcare-13-02898-t003:** Group comparison (trained vs. not trained clinicians) regarding positive and negative perception of death (Mann—Whitney U test results).

Variable	Total N	Mann–Whitney U	Wilcoxon W	Standard Error	Standardized Test Statistic (Z)	Asymptotic Significance (Two-Tailed)
Burnout	138	2339.000	6080.000	226.982	0.454	0.650
Professional development	138	1859.000	5600.000	226.757	−1.663	0.096
Burnout	138	2339.000	6080.000	226.982	0.454	0.650

## Data Availability

The original contributions presented in this study are included in the article. Further inquiries can be directed to the corresponding author.

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
