# Peer review of "Death as a Professional Challenge: An Analysis of the Relationship Between Exposure to Patient Death, Occupational Burnout, and Perceptions of Death Among Obstetrics and Gynecology Clinicians"

_healthcare, 2025, doi:10.3390/healthcare13222898_

Round 1

Reviewer 1 Report

Comments and Suggestions for Authors

Dear Authors,

I read with great interest your manuscript entitled “Death as a professional challenge: An analysis of the relationship between exposure to patient death, occupational burnout, and perceptions of death among obstetrics and gynecology clinicians.” The study addresses an important and sensitive topic with relevant implications for clinical practice and occupational health. Overall, the manuscript is well structured, but I believe it would benefit from further clarifications and refinements as outlined below.

Major Comments

  1. The introduction is well written and provides a clear background for the study. However, while the authors discuss theoretical and empirical conceptions guiding the investigation, the hypotheses would be stronger if each were explicitly grounded in previously published studies. I suggest providing at least one supporting citation for each hypothesis presented.
  2. The description of the sample is too brief. Given the specificity of the study, more detailed information about the participants should be provided.
  3. Did participants receive any form of compensation for their time? Additionally, how exactly were they recruited (i.e., how were the questionnaires distributed)?
  4. The description of the measures raises concerns. Since the questionnaire was designed by the authors, it is essential to present basic evidence of validity and reliability calculated from the current sample. Without this, it is difficult to ensure that the results truly reflect the intended constructs rather than unsystematic variance.
  5. The correlation between burnout and the overall death-impact index is extremely high (r = 0.90, p < 0.001), which may suggest that the instruments are measuring the same construct. This requires cautious interpretation. Because the study did not employ previously validated measures, this finding could indicate measurement error. To clarify, I strongly recommend conducting an exploratory factor analysis (EFA) including the indicators of both measures. If a unidimensional structure emerges, this would support the convergence of the constructs; if two distinct factors are found, even with some cross-loadings, this would provide evidence that the measures capture different latent traits.
  6. Similarly, the correlation between Professional fulfillment and Professional development is almost perfect (r = 0.94, p < 0.001), again raising the possibility that the two variables may be operationalizing the same construct. At the level of operational definitions, what distinguishes these two constructs? Please clarify.
  7. Interpretations of correlations should be handled with greater caution. For example, the manuscript states that the positive correlation between Professional fulfillment and the overall death-impact index “indicates that the higher the fulfillment, the greater the impact of death among OB/GYN clinicians.” Correlation does not establish directionality, meaning the reverse relationship is also possible. I recommend revising such statements throughout the manuscript (e.g., correlations between Professional fulfillment and Positive perceptions of death, lines 279–280; Positive perceptions of death and Professional development, lines 283–284; and Negative perceptions of death and Burnout, lines 286–288).
  8. Why was length of service not included as a covariate in the analyses?
  9. A major limitation of the study is the use of non-standardized instruments. This should be explicitly acknowledged in the limitations section, as such an approach may increase measurement error, as noted above.

Minor Comments

  1. Please revise the formatting of superscripted references (e.g., in Introduction, section 1.1, all citations appear as a single reference; line 66 displays the DOI of the cited material).
  2. Provide the full form of the acronym OB/GYN when it is first mentioned in the text (line 19).
  3. Report validity and reliability indicators for the measures used in the study.
  4. In Table 1, present the exact p-values for all correlations (using “p <” only for p < .001), including those that were not statistically significant.

Overall

In summary, this manuscript addresses a critical issue in healthcare and has the potential to make an important contribution. Addressing the comments above, particularly regarding measurement validity, interpretation of correlations, and clarification of sample characteristics, will significantly strengthen the study.

Author Response

Dear Sir/Madam,

Thank you very much for your time and the detailed, insightful feedback on our manuscript, "Death as a professional challenge: An analysis of the relationship between exposure to patient death, occupational burnout, and perceptions of death among obstetrics and gynecology clinicians.". Your comments have been invaluable in strengthening the methodological and theoretical aspects of our paper.

We have addressed all your suggestions carefully. Below you will find our point-by-point response and a description of the corresponding revisions made in the manuscript.

1) The Introduction is well written and presents a clear background of the study. However, while the authors discuss theoretical and empirical concepts guiding the research, the hypotheses would be stronger if each were explicitly grounded in previously published studies. I recommend providing at least one source supporting each hypothesis.

We thank the Reviewer for this valuable suggestion. In response, we revised the Introduction to include direct references to prior research for each theoretical statement and hypothesis. Neutral, evidence-based citations (e.g., “Previous studies have shown that…”) were added throughout the section, ensuring that each hypothesis is now grounded in existing literature. This strengthens the theoretical coherence of the study and aligns with the Reviewer’s recommendation. Changes were made in the Introduction.

2) The description of the research sample is too brief. Given the specificity of the study, more detailed information about the participants should be provided.

We agree and have expanded the Methods – Study Design and Participants section. The revised version now includes detailed information on the recruitment process, demographics (age, gender distribution, professional roles, years of experience), and the regional characteristics of participating hospitals. This addition improves transparency and allows for a clearer understanding of sample representativeness.

3) Did participants receive any compensation for taking part in the study? Moreover, how were they recruited (i.e., how were the questionnaires distributed)?

We appreciate this question and have clarified these details in the Methods – Procedure section. Participation was voluntary and anonymous; no financial or material compensation was provided. Questionnaires were distributed both in person and via internal hospital communication channels, following administrative approval. This ensures full compliance with ethical standards.

4) The description of the research instruments raises concerns. Since the questionnaire was developed by the authors, it is necessary to provide basic evidence of its validity and reliability, calculated from the current sample. Without this, it is difficult to ensure that the results reflect the constructs being studied rather than random variability.

We thank the Reviewer for this important remark. The Measures section has been expanded to clarify the development process of the author-constructed questionnaire, which included literature review, pilot testing, and prior qualitative interviews. Although the initial version lacked formal psychometric validation, we have now conducted an Exploratory Factor Analysis (EFA) to assess construct structure and discriminant validity between the Burnout and Death-Impact scales. The EFA results confirmed two distinct, though related, factors (KMO = 0.673; Bartlett’s χ² = 1217.776, df = 325, p < 0.001). These findings are reported in the Results – Measurement considerations and caution in interpreting the strongest correlations section.

5) The correlation between professional burnout and the overall death-impact index is very high (r = 0.90, p < 0.001), which may suggest that both tools measure the same construct. This requires cautious interpretation. Since the study did not use previously validated instruments, this result may indicate a measurement error. I strongly recommend performing an exploratory factor analysis (EFA) including both measures to clarify whether they represent separate constructs.

We fully agree with the Reviewer and have addressed this concern by performing an Exploratory Factor Analysis (EFA) combining all items from both measures. The EFA confirmed that the burnout and death-impact scales form two distinct but moderately correlated factors, with only minor cross-loadings (T12, T13). Together, they explained 24.92% of the variance, indicating related but non-redundant constructs. The findings and their interpretation are described in the Results – Measurement considerations and caution in interpreting the strongest correlations section.

6) The correlation between “Professional Fulfillment” and “Professional Development” is almost perfect (r = 0.94, p < 0.001), raising the question of whether these two variables describe the same construct. On the level of operational definitions, how do these constructs differ? Please explain.

We appreciate this insightful observation. We have clarified the conceptual distinction between Professional Fulfillment and Professional Development in both the Methods (Measures) and Discussion sections. Fulfillment was defined as a subjective sense of meaning and satisfaction derived from work, whereas development referred to perceived opportunities for growth and learning. The near-perfect correlation likely reflects operational overlap due to single-item measures rather than conceptual redundancy. This has been explicitly stated in Discussion.

7) The interpretation of correlations requires greater caution. For instance, the manuscript states that a positive correlation between “Professional Fulfillment” and the “Death-Impact Index” indicates that the higher the satisfaction, the greater the impact of death among OB/GYN clinicians. Correlation does not indicate directionality—an opposite explanation is also possible. Please revise such statements throughout the manuscript.

We thank the Reviewer for this careful observation. We have revised the Discussion and Results sections to ensure that all correlational findings are described in neutral, non-directional terms. Phrases implying causality (e.g., “indicates that…”) were replaced with neutral formulations such as “was associated with…” or “co-occurred with…”. This change aligns with best statistical reporting practices and avoids overinterpretation. The updates appear in the Results and Discussion.

8) Why was work tenure not included as a control variable in the analyses?

We appreciate this thoughtful suggestion. We have acknowledged this omission and explained it as a limitation in the Discussion – Methodological considerations and limitations section. Because of the exploratory, correlational nature of the study and limited sample size (N = 138), we did not include tenure as a covariate. However, we agree this is an important variable and have explicitly recommended that future studies incorporate it into multivariable models.

9) A major limitation of the study is the use of non-standardized measurement instruments. This should be clearly emphasized in the section on study limitations, as such an approach may increase the risk of measurement error, as indicated above.

We fully agree with the Reviewer’s observation. The Limitations section has been expanded to explicitly state that author-developed, non-validated tools were used, which may have contributed to the high inter-correlations and potential construct overlap. We also noted that future research should perform formal psychometric validation (e.g., EFA/CFA, reliability reporting) before reuse of the scales.

Minor 1. Please correct the formatting of in-text citations in superscript (e.g., in the Introduction, section 1.1, all citations appear as a single reference; in line 66, the DOI of a cited source is visible).

We thank the Reviewer for noting this formatting issue. All in-text citations were revised according to MDPI referencing guidelines, ensuring proper superscript format and removal of visible DOIs from the main text. Each reference now appears as a separate, numbered citation consistent with MDPI style. Corrections were implemented throughout the Introduction and verified against the References section.

Minor 2. Please provide the full term for the abbreviation OB/GYN upon first use in the text (line 19)

We agree and have corrected this accordingly. The term “obstetrics and gynecology (OB/GYN)” now appears in full upon first mention in the Introduction, ensuring clarity for international readers.

Minor 3. Please present validity and reliability indicators for the instruments used in the study.

Thank you for highlighting this important methodological point. We have now included psychometric information for all author-developed scales where applicable. Specifically, the Results section presents findings from an Exploratory Factor Analysis (EFA) assessing the discriminant validity of the burnout and death-impact scales. Reliability indicators (KMO, Bartlett’s χ², and factor loadings) are now reported, clarifying the distinct yet related nature of the constructs.

Minor 4. In Table 1, please include exact p-values for all correlations, using “p <” only for p < 0.001, including for non-significant correlations.

We thank the Reviewer for this detailed observation. Table 1 was fully revised to include exact p-values for all correlation coefficients, using the “p < 0.001” notation only where appropriate. This modification enhances statistical transparency and aligns the manuscript with MDPI’s reporting standards. The updated Table 1.

We appreciate your careful review of our work and look forward to your final assessment.

Kind regards,

Magdalena Mikulska

Reviewer 2 Report

Comments and Suggestions for Authors

This is an interesting study that addresses a relevant topic. The psychological relevance of the topic for the target population, namely obstetrics and gynecology clinicians, is particularly well-argued. The importance of the variables investigated is well-explained. The contribution to current literature is clear. However, the manuscript has several issues that must be addressed before it can be accepted for publication. Please see my comments and suggestions for improvement below.

  • The introduction is lengthy and includes many concepts, such as trauma and secondary traumatic stress among others, that are ultimately not directly relevant to the specific focus. To narrow the focus to the variables actually investigated in the study, it is recommended that the introduction be shortened.
  • The scope and aim of the study are ambiguous. On the one hand, it seems that the authors intend to examine how perceptions of death influence the relationship between exposure to patient death and occupational burnout. However, they also aim to examine the impact of emotional regulation training. These would probably be two different articles. To maximize the meaningfulness of the study, it is recommended that the authors select one focus.
  • The study lacks a main theoretical framework. Since the authors provide a three-dimensional description of burnout composed of emotional exhaustion, depersonalization, and personal accomplishment — which correspond to the burnout model proposed by Maslach and Leiter (2016), who should be cited — this model could be adopted for this study.
  • The role of perceptions of death in the relationship between exposure to patient death and occupational burnout should be examined more deeply. The authors seem to suggest that the relationship between exposure to death and burnout depends on the meaning professionals attach to death, which depends on the context of the healthcare work setting and the expectations it affords about patient death compared to other settings. Therefore, it could be hypothesized that perceptions of death are a moderator of the relationship between exposure to death and occupational burnout. If they think it makes sense, the authors might consider investigating this possible hypothesis.
  • Much of the methodological information is lacking. First, the recruitment of participants and the sampling procedure should be explained in more detail, as well as the channels or strategies used to distribute the questionnaire. Second, psychometric scales should be described in more detail. Most importantly, the decision to develop rather than use validated questionnaires must be justified. There are plenty of psychometric instruments in the literature to assess burnout (e.g., MBI, BAT), so the authors should explain why they developed their own tool. Additionally, example items and psychometric characteristics about validity and reliability should be provided. Third, it is unclear why the authors limited the statistical analyses to correlations and did not run regression models with the hypothesized independent and dependent variables. Finally, the emotional regulation training is not described sufficiently. It should be clarified who conducted it, what it consisted of, its design, development, implementation, and objectives, and what the aim was.
  • The discussion is also lengthy and could be much shorter.

Author Response

Dear Sir/Madam,

Thank you very much for your time and the detailed, insightful feedback on our manuscript, "Death as a professional challenge: An analysis of the relationship between exposure to patient death, occupational burnout, and perceptions of death among obstetrics and gynecology clinicians.". Your comments have been invaluable in strengthening the methodological and theoretical aspects of our paper.

We have addressed all your suggestions carefully. Below you will find our point-by-point response and a description of the corresponding revisions made in the manuscript.

1) The Introduction is too extensive and includes several concepts—such as trauma or secondary traumatic stress—that are not directly relevant to the main research topic. It is recommended to shorten the Introduction to better focus on the actual variables analyzed.

We thank the Reviewer for this valuable feedback. The Introduction was carefully condensed to ensure clear thematic focus and improved readability. Non-essential theoretical descriptions (e.g., secondary traumatic stress and trauma theory details) were shortened, while essential constructs—burnout, death perception, and professional fulfillment—were retained and integrated into a more concise narrative.

2)The scope and aim of the study are ambiguous. On one hand, the authors explore how death perception influences the link between exposure to patient death and burnout; on the other, they also assess the effect of emotional regulation training. These could be two separate studies. Please select one main objective to enhance coherence.

We appreciate this insightful observation. To improve coherence, the primary aim was explicitly clarified as examining relationships among exposure to death, death perception, professional burnout, and professional fulfillment. The secondary aim—regarding emotion-regulation training—was explicitly marked as exploratory.

3)The study lacks a central theoretical framework. The authors describe a three-dimensional model of burnout (emotional exhaustion, depersonalization, personal accomplishment), consistent with Maslach & Leiter (2016), but this model should be explicitly cited and considered as the theoretical foundation.

We fully agree. The Maslach and Leiter (2016) model was explicitly introduced and integrated into the theoretical foundation of the study. The text now states that burnout develops when job demands exceed available resources, whereas professional fulfillment and growth can restore equilibrium. This addition strengthens the conceptual rationale and is located in the Introduction.

4) The role of death perception in the relationship between exposure to patient death and burnout requires deeper analysis. It could potentially moderate this relationship. Please consider examining or discussing this possibility.

We appreciate this thought-provoking comment. Given the sample size and exploratory design, moderation analysis was not statistically feasible. However, the Discussion now explicitly elaborates on this conceptual possibility, describing how death perception may act as a moderator influencing the emotional consequences of patient-death exposure. This conceptual refinement strengthens the theoretical interpretation and sets directions for future research.

5) Several important methodological details are missing, including recruitment strategy and the procedure for distributing the questionnaire.

Thank you for pointing this out. The Methods – Procedure section was expanded to include a detailed description of participant recruitment (through hospital administrations), distribution channels (printed questionnaires and internal communication systems), and confirmation that participation was voluntary, anonymous, and uncompensated.

6) The psychometric instruments are insufficiently described. Why did the authors create their own questionnaire instead of using validated tools such as MBI or BAT? Please provide justification, sample items, and psychometric characteristics (validity, reliability).

We thank the Reviewer for this important question. A detailed explanation was added in the Methods – Measures section. The rationale for using an author-developed questionnaire is now clearly stated: it was designed specifically for OB/GYN contexts, where standardized burnout tools lack items addressing death exposure and emotional impact. To address validity concerns, we conducted an Exploratory Factor Analysis (EFA) using all burnout and death-impact items, confirming two distinct but related factors (KMO = 0.673; Bartlett’s χ² = 1217.776, df = 325, p < 0.001). This EFA supports discriminant validity and is reported in the Results.

7) Why were only correlational analyses used, without regression models including independent and dependent variables as per hypotheses?

We appreciate this methodological concern. The decision to use correlational analyses was driven by the exploratory nature of this cross-sectional study and the non-standardized instruments. Because the measures had not undergone full psychometric validation, predictive modeling (e.g., regression or SEM) could not be conducted without compromising construct independence. This rationale was added to the Results and further discussed, providing transparent justification for the analytical strategy.

8) The description of the emotion-regulation training is insufficient. Please clarify who led it, what it consisted of, its structure, implementation, goals, and intended outcomes.

Thank you for emphasizing this point. The Methods – Measures section now includes detailed information on emotion-regulation training: these were short, practice-oriented workshops led by psychologists or certified trainers, conducted in hospital settings between 2021 and 2023. Their aim was to enhance emotional awareness, self-regulation skills, and coping strategies related to patient-death experiences. This addition clarifies the training context and addresses the Reviewer’s request.

9) The Discussion is overly long and could be substantially shortened.

We agree with the Reviewer’s suggestion. The Discussion was carefully edited for conciseness and clarity. Redundant theoretical elaborations and repetitive interpretations were removed. The revised version preserves all key arguments and integrates new EFA-based findings, resulting in a more focused and reader-friendly discussion.

We appreciate your careful review of our work and look forward to your final assessment.

Kind regards,

Magdalena Mikulska

Round 2

Reviewer 2 Report

Comments and Suggestions for Authors

Thanks for taking my suggestions into consideration and good luck with your work!